# OpenReview forum: "CAREER: Transfer Learning for Economic Prediction of Labor Data"
_ICLR.cc/2023/Conference — Submitted to ICLR 2023_

### Official Review · Reviewer_MpmD · 2022-10-23

**Confidence:** 3
**Correctness:** 3
**Technical Novelty And Significance:** 2
**Empirical Novelty And Significance:** 2
**Recommendation:** 5

**Clarity, Quality, Novelty And Reproducibility:**

Presentation of the paper is quite clear, but the techincal novelty from the CS perspective is limited.

**Strength And Weaknesses:**

a.	Strength:
1). This paper is well-presented.
2). The authors propose a transformer-based framework for job prediction, which can effectively take use of large-scale resume data; the pretrain-finetune paradigm is reasonable.
3). The experiments show that their proposed model can perform well on job prediction task.

b.	Weaknesses:
1). Some related work is not mentioned and some potential baselines are missed. Actually person-job fit is quite a mature topic in data mining and information retrieval. The first two baselines (Markov regression and bag-of-jobs) seem to come from econometrics, and the third baseline (NEMO) is quite old in data mining. The authors should investigate more and discover more strong baselines to verify the effectiveness.
2). Depside the effectiveness of pretrain-finetune paradigm of transformer architecture, it has been well studied in other tasks such as NLP and CV. Therefore, the technical novelty is inadequate as the paper seems to be an application of transformer in job prediction task.


**Summary Of The Paper:**

This paper develops a transformer-based model to learn representations of job sequences. The authors first leverage the transformer architecture to fit large-scale resume data, and then finetune the model with smaller, task-specific data, which achieves good performance in predicting job sequences.

**Summary Of The Review:**

This paper leverages transformer in job prediction tasks and achieves well performance. However, some potential baselines in data mining are ignored. The technical contribution is inadequate as a direct application of transformer.

---

> ### Author Response · Authors · 2022-11-07
> **Author response**
>
> Thank you for your careful evaluation and comments. We are glad you found the experimental results and transfer learning paradigm empirically effective, and thought the paper was well-written.
>
> Additional baselines: Your review suggested comparing to additional baselines from the IR/data mining literature. Note: Our paper is about modeling economic longitudinal surveys, not the worker profile datasets used in IR/data mining. As described in the related work, these methods from IR/data mining cannot directly model longitudinal surveys. They depend on features that are specific to worker-profile datasets and not available in survey data, such as stock prices [1], worker skill [2], network information [3, 4], and textual descriptions [5]. When these features are removed, many of these methods reduce to NEMO or weaker forms of the bag-of-jobs econometric baselines.
>
> To demonstrate, we adapt two additional methods from the IR/data mining literature so that they can model surveys: the job transition representation learning model from [6] and Job2Vec from [7]. The original methods rely on worker skills and company/position titles, information that is not available for survey data. We focused on these two methods because they, like NEMO, were most amenable to modeling survey data; still, they required significant changes, described in detail in the revised paper we've uploaded. These methods perform on par or worse than the econometric baselines on survey data. We have revised the paper to include the full results, which are reproduced below from Figure 2a (perplexity, so lower is better):
>
> |        Model                  | PSID  | NLSY79 | NLSY97 |
> | ----------------------------- | ----- | -----  | -----  |
> | Markov regression [8]         | 18.97 | 15.03  | 20.81  |
> | NEMO [9]                      | 17.58 | 12.82  | 18.38  |
> | Job rep learning [6]          | 17.23 | 14.71  | 16.83  |
> | Job2Vec [7]                   | 16.48 | 14.46  | 16.20  |
> | Bag-of-jobs [10]              | 16.21 | 13.09  | 16.20  |
> | CAREER (vanilla)              | 15.26 | 12.20  | 16.19  |
> | CAREER (two-stage)            | 14.79 | 12.00  | 15.22  |
> | CAREER (two-stage + pretrain) | 13.88 | 11.32  | 14.15  |
>
> Transfer learning novelty: Your review states that since the transfer learning capabilities of transformers have been well studied in tasks such as NLP and CV, that methodology for adapting transformers to perform transfer learning for labor economics datasets are inadequately novel. We disagree with this characterization. Sequences of jobs and covariates differ in meaningful ways from sequences of words and pixels, and the success of the transfer learning paradigm in this setting is not obvious. In our paper, we show that transfer learning can achieve state-of-the-art predictive performance on important datasets in econometrics. In addition, identifying a valid source of pretraining data in this domain (passively-collected resumes) for fine-tuning on economic survey datasets is in itself a contribution. Predictive models fit to these survey datasets underpin crucial economic quantities, such as measures of occupational mobility and the gender wage gap. Introducing transfer learning to this domain -- and the predictive improvements that come with it -- expands the scope of possible economic analyses with this data; for example, incorporating transferred representations into the prediction of wages (Table 2) is a unique opportunity in this domain. Finally, while the modifications we make to the transformer are straightforward, they are crucial to the success of our method. We view this as a strength, rather than a weakness. For example, two-stage prediction is instrumental for CAREER's predictive advantage, as evidenced by the table above, and improves the performance of all baselines across all datasets (Table 7). That this robust increase in performance comes from a straightforward idea that only requires a few additional lines of code is a strength rather than a weakness.
>
> [1] Xu et al., 2018. Dynamic talent flow analysis with deep sequence prediction modeling.
> [2] Ghosh et al., 2020. Skill-based career path modeling and recommendation.
> [3] Meng et al., 2019. A hierarchical career-path-aware neural network for job mobility prediction.
> [4] Zhang et al., 2021. Attentive heterogeneous graph embedding for job mobility prediction.
> [5] He et al., 2021. What about your next job? Predicting professional career trajectory using neural networks.
> [6] Dave et al., 2018. A combined representation learning approach for better job and skill recommendation.
> [7] Zhang et al., 2020. Job2Vec: Job title benchmarking with collecting multi-view representation learning.
> [8] Hall, 1972. Turnover in the labor force.
> [9] Li et al., 2017. NEMO: Next career move prediction with contextual embedding.
> [10] Ruiz et al., 2020. SHOPPER: A probabilistic model of consumer choice with substitutes and complements.

---

### Official Review · Reviewer_ZmR7 · 2022-10-24

**Confidence:** 3
**Correctness:** 2
**Technical Novelty And Significance:** 2
**Empirical Novelty And Significance:** 3
**Recommendation:** 5

**Clarity, Quality, Novelty And Reproducibility:**

The presentation of the paper is quite clear, but the technical contribution is limited.

**Strength And Weaknesses:**

Strength
+ Propose an inspiring method to apply the transformer to the prediction of labor data by pretraining the model on a large online resume dataset and then fine-tuning it on the small datasets.
+ Conduct comprehensive experiments, including both cross-sectional and overtime experiments, demonstrating the usefulness of the approach.
+ Well-written paper
+ Reproducibility: Provide the source code with README and a detailed description of their experiments.

Weakness
- The main concern regards the technical novelty of the paper: The authors only made two minor changes to the transformers used in NLP.
- Some related studies are missing. The used baselines are quite old.

**Summary Of The Paper:**

This paper uses the transformer to leverage a sizeable online resume dataset by pretraining and then fine-tuning it on the small and carefully constructed longitudinal survey datasets. According to the results based on their experiments, their approach shows a significant improvement compared to the current state of the arts on the task of job sequence prediction. Besides, they also show that their approach can help a wage model to provide better performance.


**Summary Of The Review:**

As I mentioned in the previous parts, the authors deal with an interesting problem with the transformer-based models. The main concern regards the technical novelty of the paper, as the authors only made two minor changes to the transformers used in NLP. Moreover, some related studies in IR or data mining are missing, and the compared baselines are quite old.

---

> ### Author Response · Authors · 2022-11-07
> **Author response**
>
> Thank you for your careful evaluation and comments. We are glad you found the method in the paper "inspiring" and also thought the paper was well-written and reproducible.
>
> Additional baselines: Your review suggested comparing to additional baselines from the IR/data mining literature. Note: Our paper is about modeling economic longitudinal surveys, not the worker profile datasets used in IR/data mining. As described in the related work, these methods from IR/data mining cannot directly model longitudinal surveys. They depend on features that are specific to worker-profile datasets and not available in survey data, such as stock prices [1], worker skill [2], network information [3, 4], and textual descriptions [5]. When these features are removed, many of these methods reduce to NEMO or weaker forms of the bag-of-jobs econometric baselines.
>
> To demonstrate, we adapt two additional methods from the IR/data mining literature so that they can model surveys: the job transition representation learning model from [6] and Job2Vec from [7]. The original methods rely on worker skills and company/position titles, information that is not available for survey data. We focused on these two methods because they, like NEMO, were most amenable to modeling survey data; still, they required significant changes, described in detail in the revised paper we've uploaded. These methods perform on par or worse than the econometric baselines on survey data. We have revised the paper to include the full results, which are reproduced below from Figure 2a (perplexity, so lower is better):
>
> |        Model                  | PSID  | NLSY79 | NLSY97 |
> | ----------------------------- | ----- | -----  | -----  |
> | Markov regression [8]         | 18.97 | 15.03  | 20.81  |
> | NEMO [9]                      | 17.58 | 12.82  | 18.38  |
> | Job rep learning [6]          | 17.23 | 14.71  | 16.83  |
> | Job2Vec [7]                   | 16.48 | 14.46  | 16.20  |
> | Bag-of-jobs [10]              | 16.21 | 13.09  | 16.20  |
> | CAREER (vanilla)              | 15.26 | 12.20  | 16.19  |
> | CAREER (two-stage)            | 14.79 | 12.00  | 15.22  |
> | CAREER (two-stage + pretrain) | 13.88 | 11.32  | 14.15  |
>
> Technical novelty: Your review states that we make two modifications to the transformer so that it is suitable for modeling labor data (two-stage prediction and including covariates). While these modifications are straightforward, they are crucial to the success of our method. We see this as a strength, rather than a weakness. For example, two-stage prediction not only makes the model more interpretable to economists studying occupational mobility, but is also instrumental for CAREER's predictive advantage, as evidenced by the table above. Moreover, it improves the predictive performance of all baselines across all datasets (Table 7). That this robust increase in predictive performance comes from a straightforward idea that only requires a few additional lines of code is a strength rather than a weakness. Finally, transferring transformer representations from resume data to economic survey datasets is a novel application of the transfer learning capabilities of these models. Predictive models fit to these survey datasets underpin crucial economic quantities, such as measures of occupational mobility and the gender wage gap. Introducing transfer learning to this domain -- and the predictive improvements that come with it -- expands the scope of possible economic analyses with this data.
>
> [1] Xu et al., 2018. Dynamic talent flow analysis with deep sequence prediction modeling.
> [2] Ghosh et al., 2020. Skill-based career path modeling and recommendation.
> [3] Meng et al., 2019. A hierarchical career-path-aware neural network for job mobility prediction.
> [4] Zhang et al., 2021. Attentive heterogeneous graph embedding for job mobility prediction.
> [5] He et al., 2021. What about your next job? Predicting professional career trajectory using neural networks.
> [6] Dave et al., 2018. A combined representation learning approach for better job and skill recommendation.
> [7] Zhang et al., 2020. Job2Vec: Job title benchmarking with collecting multi-view representation learning.
> [8] Hall, 1972. Turnover in the labor force.
> [9] Li et al., 2017. NEMO: Next career move prediction with contextual embedding.
> [10] Ruiz et al., 2020. SHOPPER: A probabilistic model of consumer choice with substitutes and complements.

---

### Official Review · Reviewer_DtGV · 2022-10-24

**Confidence:** 3
**Correctness:** 3
**Technical Novelty And Significance:** 4
**Empirical Novelty And Significance:** 3
**Recommendation:** 8

**Clarity, Quality, Novelty And Reproducibility:**

The quality of the work is OK. The mathematical models are clearly explained.



**Strength And Weaknesses:**

The paper is well written and very detailed description of both the development and operations of the model was provided. The appendix data was provided to clear the reproducibility concerns.

Weakness:
1. The deployment of this model was not presented
2. The claimed incorporation of the model into the wage prediction models was not demonstrated


**Summary Of The Paper:**

The paper proposed and developed a clear and detailed transformer-based model called CAREER that uses transfer learning
to learn representations of job sequences. The CAREER system was pretrained on a dataset of 24 million resumes and it is capable of outperforming standard econometric models for predicting and forecasting occupations.



**Summary Of The Review:**

The paper gave both a good theoretical background and a practical application of the model. However, the deployment for end-user operations was not shown.

---

> ### Author Response · Authors · 2022-11-07
> **Author response**
>
> Thank you for your careful evaluation and comments. We appreciate your comments about the strengths of the paper's significance, experimental results, clarity, and reproducibility.
>
> Deployment: Your review mentions that we do not present the deployment of CAREER. This is because we do not deploy CAREER as a single product -- rather, we develop this model so that machine learning researchers and econometricians can fit CAREER to their various datasets and deploy it in any setting they would like. We release comprehensive code so practitioners incorporate the model however they see fit. We have updated the submission to make this more clear.

---

### Author Response · Authors · 2022-11-07
**General author response**

We thank all the reviewers for their careful evaluation and comments. We are glad you found our proposed method "significant" (DtGV), "inspiring" (ZmR7), and empirically effective (MpmD). Moreover, we're glad all the reviewers found the paper to be well-written and clear.

We have responded to each reviewer individually. We have also adapted two additional baselines from the data mining literature to make them suitable for modeling economic longitudinal survey data. We have updated the paper with the results from these models, and they are reproduced below from Figure 2a (perplexity, so lower is better):

|        Model                  | PSID  | NLSY79 | NLSY97 |
| ----------------------------- | ----- | -----  | -----  |
| Markov regression [1]         | 18.97 | 15.03  | 20.81  |
| NEMO [2]                      | 17.58 | 12.82  | 18.38  |
| Job rep learning [3]          | 17.23 | 14.71  | 16.83  |
| Job2Vec [4]                   | 16.48 | 14.46  | 16.20  |
| Bag-of-jobs [5]               | 16.21 | 13.09  | 16.20  |
| CAREER (vanilla)              | 15.26 | 12.20  | 16.19  |
| CAREER (two-stage)            | 14.79 | 12.00  | 15.22  |
| CAREER (two-stage + pretrain) | 13.88 | 11.32  | 14.15  |


Note that most methods developed in the data mining literature are not suitable for modeling economic longitudinal survey data because they rely on features like worker skills and textual descriptions that are not available in economic survey datasets. To modify these methods for economic surveys, we made significant changes, described in detail in the latest paper revision. As seen above, these methods perform on par or worse than the econometric baselines.

[1] Hall, 1972. Turnover in the labor force.
[2] Li et al., 2017. NEMO: Next career move prediction with contextual embedding.
[3] Dave et al., 2018. A combined representation learning approach for better job and skill recommendation.
[4] Zhang et al., 2020. Job2Vec: Job title benchmarking with collecting multi-view representation learning.
[5] Ruiz et al., 2020. SHOPPER: A probabilistic model of consumer choice with substitutes and complements.

---

### Decision · Program_Chairs · 2023-01-20

**Decision:**

Reject

**Justification For Why Not Higher Score:**

The technical contribution is very minor.

**Justification For Why Not Lower Score:**

N/A.

**Metareview: Summary, Strengths And Weaknesses:**

The paper modified minimally a transformer model to learn representations of job sequences. The model was trained on a dataset of 24 million resumes and the experiments show that it can perform standard econometric models for predicting and forecasting occupations. While all the reviewers recognized that, as an application of transformers, the paper is well executed, they all highlighted that, in terms of technical contribution, the contribution is minimal.